
# Evolution of source attributed organic aerosols and gases in a megacity of central China

Siyuan Li [1], Dantong Liu [1,*], Shaofei Kong [2,*], Yangzhou Wu [1], Kang Hu [1], Huang Zheng [2], Yi Cheng [2], Shurui Zheng [2], Xiaotong Jiang [1], Shuo Ding [1], Dawei Hu [3], Quan Liu [4], Ping Tian [5], Delong Zhao [5], Jiujiang Sheng [5]

[1] Department of Atmospheric Sciences, School of Earth Sciences, Zhejiang University, Hangzhou 310027, China

[2] Department of Atmospheric Sciences, School of Environmental Studies, China University of Geosciences, Wuhan, 430074, China

[3] Centre for Atmospheric Sciences, School of Earth and Environmental Sciences, University of Manchester, Manchester M13 9PL, UK

[4] State Key Laboratory of Severe Weather & Key Laboratory of Atmospheric Chemistry of CMA, Chinese Academy of Meteorological Sciences, Beijing 100081, China

[5] Beijing Key Laboratory of Cloud, Precipitation and Atmospheric Water Resources, Beijing Meteorological Service, Beijing 100089, China; Field Experiment Base of Cloud and Precipitation Research in North China, China Meteorological Administration, Beijing 100089, China.

*Correspondence to*: Dantong Liu (dantongliu@zju.edu.cn); Shaofei Kong (kongshaofei@cug.edu.cn)

**Abstract.** The secondary production of the oxygenated organic aerosol (OOA) impacts air quality, climate and human health. The importance of various sources in contributing to the OOA loading and associated different aging mechanisms remains to be elucidated. Here we present concurrent observation and factorization analysis on the mass spectra of organic aerosol (OA) by a high-resolution aerosol mass spectrometer and volatile organic compounds (VOCs) by a proton-transfer-reaction mass spectrometer in Wuhan, a megacity in central China during autumntime. The full mass spectra of organics with two principle anthropogenic sources were identified as the traffic and cooking sources, for their primary emission profiles in aerosol and gas phases, the evolutions, and their respective roles in producing OOA and secondary VOCs. Primary emissions in gas and aerosol phases both contributed to the production of OOA. The photooxidation of traffic sources from morning rush-hour caused 2.5-folds increase of OOA mass in a higher oxidation state (O/C=0.72), coproducing gas-phase carboxylic acids; while at night, cooking aerosols and VOCs (particularly acrolein and hexanal) importantly caused the nocturnal formation of oxygenated intermediate VOCs, increasing OOA mass by a factor of 1.7 (O/C=0.42). The daytime and nighttime formation of secondary aerosols as contributed by different sources, was found to be modulated by solar radiation and air moisture respectively. The environmental policy should therefore consider the primary emissions and their respective ageing mechanisms influenced by meteorological conditions.



## 1 Introduction

The transformation between gas and aerosol phase of organic species produces secondary organic aerosols (SOA) (Claeys et al., 2004; Kroll and Seinfeld, 2008; Hallquist et al., 2009; Slowik et al., 2010), which forms important global budget of aerosol loadings (Heald et al., 2005; Zhang et al., 2007), exerting climate (Poschl, 2005; Seinfeld et al., 2016) and environmental impacts (Von Schneidemesser et al., 2011; Huang et al., 2014). Given the large complexities of organic species, the yields (Goldstein and Galbally, 2007; Ortiz-Montalvo et al., 2014) and production rates (Sareen et al., 2013; Jokinen et al., 2015) of SOA from gas precursors are influenced by the diversities of source profiles (emission mass percentages among species) (Lin et al., 2012; Shrivastava et al., 2015) and environmental factors (such as radiation, air moisture and ambient temperature) (Li et al., 2021b; Wang et al., 2021). This raises challenges for source-oriented environmental policy making, before explicit understanding on the formation mechanism of SOA from different sources.

The application of detailed mass spectra of organic aerosols allows online source attribution of organic aerosol (OA) (Canagaratna et al., 2007; Ng et al., 2011), based on the factorization analysis on the mass spectra of OA which groups the covaried species from certain sources (Ulbrich et al., 2009). This factorization technique allows identification of primary sources and aged secondary sources at a receptor measurement site. The complex aerosol sources in urban environment have been commonly recognized as traffic sources mainly from vehicles on road (Zhu et al., 2021), biomass burning from open or closed combustion (Adler et al., 2011), coal combustion normally in cold season for heating (Xu et al., 2019) and more localized cooking sources (Allan et al., 2010; Zhang et al., 2020b). The secondary sources are oxygenated OA which may be from oxidizing volatile organic compounds (VOCs) following condensation (Donahue et al., 2006), or heterogenous oxidation occurring on particle phase (Claeys et al., 2004; Kroll and Seinfeld, 2008). Among these sources, SOA has been found to be the main contributor of OA mass loading (41~69%) in urban environment of East Asia (Sun et al., 2014; Hu et al., 2017), especially in warm season when primary emissions were low, along with high ambient temperature and more intensive chemical reactions (Hu et al., 2016). The formation of SOA from VOCs may experience a few reaction generations (Knote et al., 2014) and could interact with other sources of species during the process (Shrivastava et al., 2017), hereby complicating the goal of identifying the key precursors in contributing to the consequent SOA. In addition, some primary gases already have somehow low volatility and may not require a long reaction chain to become condensable, such as some primarily emitted intermediate volatility organic compounds (IVOCs) may substantially contribute to the SOA (Robinson et al., 2007; Huang et al., 2021). An understanding of source profiles of primary emissions in both gas and aerosol phases is therefore important to rule out the source-dependant production of SOA. The above necessitates the concurrent investigation on the compositions of gases and aerosols at a receptor site, along with their evolution and interaction, in order to elucidate the role of each source in contributing to SOA.

In this study, we performed online continuous measurements on the detailed mass spectra of organics concurrently on aerosol and gas phases, in a typical anthropogenically polluted region in central China, where such data had been rarely



reported. Factorization analysis is performed on the mass spectra of organics in both aerosol and gas phases, to investigate
the source-oriented gas-aerosol evolution and SOA formation in this region.

## 2 Materials and methods

### 2.1 Sampling site

The field experiment was performed in the campus of the China University of Geosciences (Wuhan) (114.40° E, 30.52°
N) during Oct.- Nov. 2019 (Hu et al., 2021). The site represents a typical residential/traffic mixed region (Fig. S1). Due to
the preparation and hosting of the $7^{th}$ CISM Military World Games during the experimental period, the government
implemented strict emission reduction measures, particularly for industrial sources. The local pollution sources were hereby
dominated by traffic and cooking sources, besides those transported from surrounding regions during some heavy pollution
events (Zheng et al., 2019; Zheng et al., 2020). The HYSPLIT model (Draxler and Hess, 1997) with a 3-hourly $1° \times 1°$ wind
field from the GDAS reanalysis products was used to obtain the 36-h backward trajectories initialized from the location of
the experiment site. Cluster analysis was performed to categorize the trajectories into three groups by minimizing the
differences in each group and maximizing the differences among groups (Moody and Galloway, 1988).

### 2.2 Instrumentation

#### 2.2.1 Measurements of mass spectra of aerosols and gases

The mass concentration and chemical composition of NR-PM$_1$ were measured by a HR-ToF-AMS (Aerodyne Research
Inc., USA), including organics, nitrate (NO$_3^-$), sulphate (SO$_4^{2-}$), chloride (Chl$^-$), and ammonium (NH$_4^+$). The detailed can be
found elsewhere (Decarlo et al., 2006). Briefly, the aerosols are dried using a diffusion dryer before entering the AMS and
through a critical orifice into a narrow beam via an aerodynamic lens. The aerosol size is determined using the flight time of
particles to the thermal vaporization and ionization chamber. Then the aerosols are successively vaporized by a heated
surface (~600 ℃), ionized by electron ionization (EI, 70 eV), and detected by a mass spectrometer detector. During this field
observation, the HR-ToF-AMS was operated under V mode with high sensitivity. The composition-dependent collection
efficiencies (Middlebrook et al., 2012) were applied, and the ionization efficiency was calibrated using 300 nm pure
ammonium nitrate (Jayne et al., 2000). Elemental analysis (EA) was also executed using the "Improved Ambient" method
(Canagaratna et al., 2015) to obtain the hydrogen-to-carbon ratio (H/C), oxygen-to-carbon ratio (O/C), and nitrogen-to-
carbon ratio (N/C).
The Proton-Transfer-Reactor Time-of-Flight Mass Spectrometer (PTR-ToF-MS 8000, Ionicon Analytik GmbH
Innsbruck, Austria) was deployed to quantify VOCs in this research. The operating and calibration of the PTR followed the
routine described previously (Cappellin et al., 2012; Bruns et al., 2016). Briefly, the PTR was operated with hydronium ions
(H$_3$O$^+$) as the reagent and with a drift tube pressure of 2.2 mbar, voltage of 600 V and temperature of 60 ℃. The ratio of the
electric field (E) and the density of the buffer gas (N) in the drift tube, which dictates the ion drift velocity in the drift tube, is



135 Td. MS transmission function was performed using a mixture of VOCs (formaldehyde, methanol, acetonitrile,
acetaldehyde, acetone, isoprene, methyl ethyl ketone, benzene, toluene, styrene, benzaldehyde, ethylbenzene, 1,3,5-
trimethylbenzene). Mass calibration was done using $H_3O^+$ (*m/z* 21.0226), $CH_3COCH_4^+$ (*m/z* 59.0490) and monoterpenes (*m/z*
137.1290). The shift in m/z is minor which ensures that the mass calibration was sufficient for all compounds. The
background measurements were performed using a dry zero air cylinder. The measurement error is described in Text S2.  A
separate reaction rate constant is applied to convert the ion signal into concentration (Bruns et al., 2016), or the default
reaction rate constant $2 \times 10^{-9}$ cm$^3$ s$^{-1}$ can be applied to all other ions (Wang et al., 2020b). The vapor saturation concentration
(equilibrium vapor pressures) ($C^*$) of each VOCs compound at 25ºC is estimated using the parameterization based on
elemental ratio and molecular weight (Fig. S2) (Li et al., 2016). A log$C^*$ (in μg m$^{-3}$) lower than 6.5 is deemed to be
intermediate VOC (IVOC). According to Fig. S2, VOC species (with *m/z* >120) are mainly identified to be IVOCs in this
study, thus the fraction of these larger molecular weight (MW) VOCs (*m/z* >120) is used to evaluate the potential influence
of IVOCs.
**2.2.2 PMF analysis on the mass spectra of OA and VOCs**
Positive matrix factorization (PMF) (Paatero and Tapper, 1994) was performed on the high-resolution organic mass
spectral matrix of OA (Ulbrich et al., 2009; Decarlo et al., 2010). In this work, m/z > 120 and isotopic ions were excluded in
PMF analysis due to the limited mass resolution and low contributions to OA loading (~5%). After a careful evaluation of
the mass spectral profiles and correlations with time series of tracers, diurnal variations, four OA factors from total OA were
identified with *f*peak = 0, including hydrocarbon-related OA (HOA), cooking OA (COA), low-oxidized oxygenated OA
(LO-OOA, OOA1), and more-oxidized oxygenated OA (MO-OOA, OOA2). The detail of PMF diagnostic was summarized
in Text S1 and Fig. S3.
The EPA PMF 5.0 model (Paatero and Tapper, 1994) was used for the source apportionment of VOC species. The 109
VOCs were used for the PMF analysis. The uncertainties from each sample were determined according to the method
detection limit (MDL) and the error fraction (%). The detail of PMF diagnostic was summarized in Text S2 and Fig. S4. Five
factors were ultimately selected, and the $Q/Q_{exp}$ ratio was 0.96 on average (Fig. S4). The rotation ambiguity was explored by
varying the *f*peak values from -3 to +3, and the results with *f*peak = 0 were selected for the lowest d$Q$ (robust), indicating the
stability of the PMF solution (Zhou et al., 2019). Most of the residuals are distributed normally, ranging from -3 and +3,
suggesting the model fit the input data well.
**2.2.3 Other measurements**
The concentration of BC particles was measured by a single particle soot photometer (SP2, DMT Inc.). The operation
and data analysis procedures of the SP2 have been described elsewhere (Schwarz et al., 2008; Liu et al., 2010). The SP2
incandescence signal was calibrated for refractory BC (rBC) mass using the Aquadag black carbon particle standards
(Acheson Inc., USA) and corrected for ambient rBC with a factor of 0.75 (Laborde et al., 2012).





The size distribution and number concentration of aerosols with a mobility diameter from 12 to 552 nm were also
measured by a scanning mobility particle sizer (SMPS, Model 3081, impactor 50% cut off at 0.677 μm; CPC model 3775 at
a flow rate 0.3 L/min). $PM_1$ mass concentration calculated based on the volume concentration measured by the SMPS agreed
well with that from the sum of compositions by the HR-ToF-AMS and SP2 (r= 0.71).
The gas-phase species including $NO_x$, $O_3$, and CO were measured in real time by a series of Wuhan Tianhong analyzers
(TH- 2001H, TH-2003H, TH-2004H, respectively). These instruments were calibrated periodically with the corresponding
standard gas to ensure the accuracy of the observation data. In addition, the meteorological parameters including temperature
(T), relative humidity  (RH), and wind speed and direction were recorded by an automatic weather station.

## 3 Results and discussion

### 3.1 Chemical compositions of $PM_1$

The time series of mass concentrations of BC, nonrefractory-$PM_1$ (NR-$PM_1$) species (i.e., OA, $SO_4^{2-}$, $NO_3^-$, $NH_4^+$, and
$Cl^-$) and their relative contributions are summarized in Fig. 1. During this field observation, the mass concentrations of NR-
$PM_1$ were in the range of 2.5 - 44.8 μg m$^{-3}$, with an average of 12.7 ± 5.7 μg m$^{-3}$, which is close to those observed in the
North China Plain in autumn 2019 (15.1 μg m$^{-3}$) (Li et al., 2021a), and was much lower than that (41.3 ± 42.7 μg m$^{-3}$)
observed in Beijing in autumn 2012 (Hu et al., 2017). The mean mass concentration of BC was 0.29 ± 0.17 μg m$^{-3}$ with the
range of 0.1 - 1.0 μg m$^{-3}$. Among all species in NR-$PM_1$, OA contributed the major (49.2 %), indicating the dominant role of
OA in autumntime of $PM_1$ pollution in this region. Inorganic aerosol accounted for 50.8 % of NR-$PM_1$ in which sulfate was
the largest contributor (21.5 %), followed by nitrate (18.5 %), ammonium (10.7 %), and chloride (0.1 %), which has similar
relative contributions to those observed in field (Chen et al., 2021). The number concentration of all particles peaked at 95 ±
38 nm in with regular growth during every field day. Fig. 1 also presents the time series variation in meteorological
parameters. During the field observation period, the average temperature and relative humidity (RH) were 19.3 ± 3.6 ℃
(11.2– 29.4 ℃) and 75.0 % ± 17.5 % (25.0 %–99.0 %), respectively. The $O_3$ concentration with an average of 30.7 ± 13.0
ppb, was highest 54.5 ppb at 15:00, likely due to the high temperatures and enhanced photochemical processing.
Fig. 1f shows the spatial distribution of aerosol optical depth (AOD) and 36 h backward trajectories during the
campaign. Cluster-1 (C1) shows the circulated air mass with shortest transport distance for the past 36 h (36% fraction); C2
is the northerly transported air mass (55% fraction), over regions with higher AOD; C3 is the most rapid transport through
the northeast China and some coastal areas. Fig. 1g shows the diurnal variations of solar radiation and RH, peaking in the
day and night respectively.

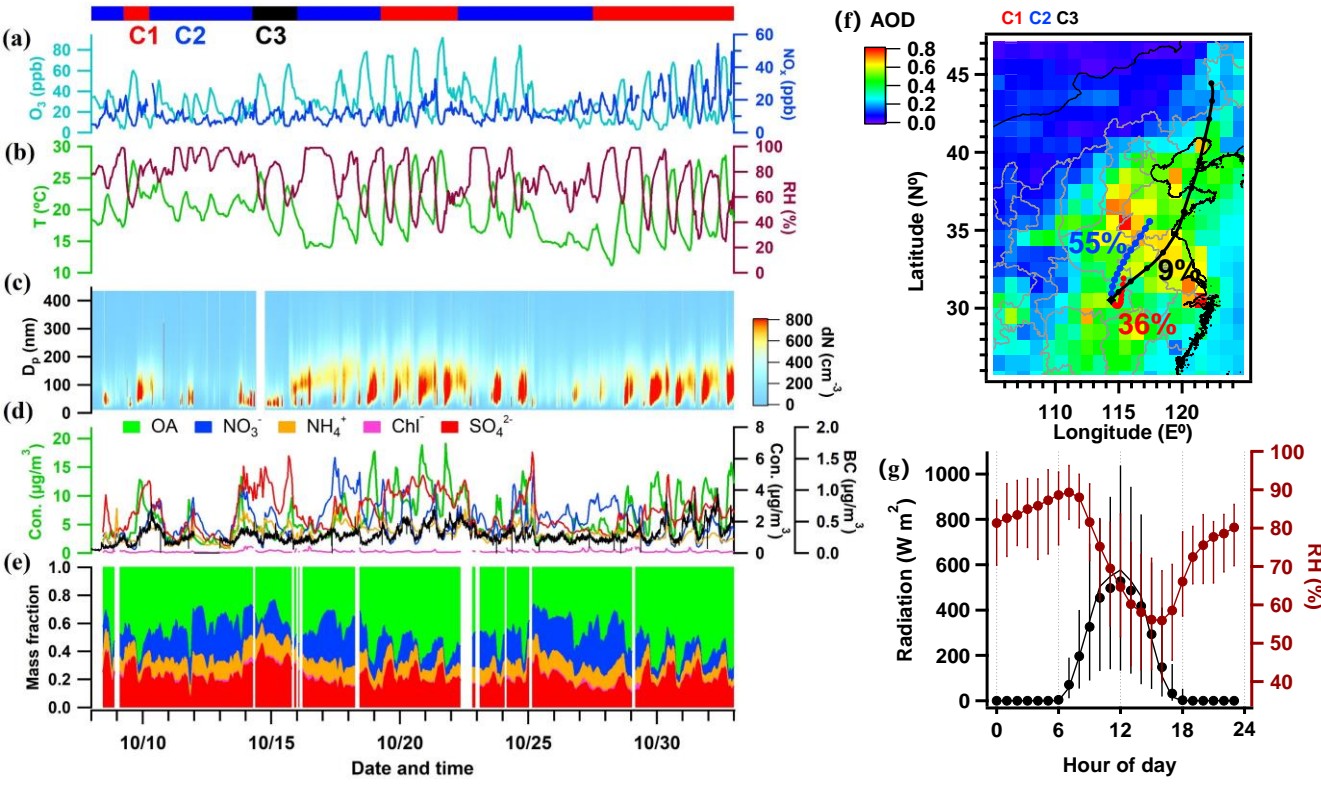

**Figure 1**: **Time series of (a) mass concentrations of O₃ and NOₓ; (b) ambient temperature (T) and relative humidity (RH); (c) number size distribution measured by the SMPS; (d) mass concentrations of key aerosol species; (e) mass fractions of chemical species in non-refractory PM₁; (f) spatial distribution of mean aerosol optical depth (AOD) during the experiment and the clustered 36 h backward trajectories; (g) diurnal profiles of direct solar radiation and RH.**

**3.2 Source attributed organic aerosol**

The resolved OA factors by the PMF analysis on the AMS measurements are shown in Fig. 2. The first factor is characterized by prominent hydrocarbon ion series of $C_nH_{2n-1}^+$ and $C_nH_{2n+1}^+$ (Fig. 2a), with a low O∕C ratio (0.24), which is generally considered to be related to the emissions of fossil fuel combustion and vehicle emissions (Huang et al., 2010; Morgan et al., 2010; N. L. et al., 2011). The time series of the factor correlated well with $NO_x$ (r = 0.71) and BC (r =0.83) (Fig. 2e and Table S1). The diurnal pattern of this factor showed peaks in the morning and afternoon rush-hour (Fig. 2i), with a major increase from 5:30, reaching a peak value of 1.5 µg m⁻³ at 8:30. The concentration gradually decreased around noontime due to boundary layer dilution until 15:00 and reached a minimum of 0.6 µg m⁻³. This spectrum also contained fragment marker for possible coal combustion OA (CCOA), i.e. $C_9H_7^+$ (*m/z* 115, r = 0.73) (Hu et al., 2013). This factor was not distinctly resolved in this dataset maybe due to the urban nature of the site, where the traffic source may have overwhelmed, due to the less significant coal combustion pollutants during the sampling period.



The second factor was characterized by $m/z$ 55 ($C_4H_7^+$, $C_3H_3O^+$) and 57 ($C_4H_9^+$, $C_3H_5O^+$), accounting for 10% and 3.5%
of the total spectrum, respectively (Fig. 2b), with the lowest O/C ratio among factors (0.16). This factor has a
$C_3H_3O^+/C_3H_5O^+$ of 3 and $C_4H_7^+/C_4H_9^+$ of 2 (1 is usually for HOA), indicating it as cooking source rather than HOA (Mohr et
al., 2012). The correlation coefficient of the COA factor and marker ion $C_6H_{10}O^+$ was 0.91, which is also similar to a
previous study (Sun et al., 2011). The diurnal pattern of this factor showed a major peaking during 18:00 - 20:00, reaching
up to 4.0 µg m$^{-3}$ on average, in addition to a smaller peak at lunch time, which corroborated the diurnal cooking activities.
Previous studies reported COA accounted for 6.5 - 30 % of the total OA in urban (Rogge et al., 1991; Lanz et al., 2008;
Allan et al., 2009; Xu et al., 2014). Here the concentrations and proportions of COA in OA were in the range of 0.5 - 4.5 µg
m$^{-3}$ and accounted for 23 % of OA on average. The cooking emission in the studying region was likely from charcoal-
grilling activities, which are popular in the surrounding areas.
Besides the two primary OA, additional two oxygenated OA (OOA) factors are identified. According to the oxidation
state, OOA was further separated into lower (OOA1) and more oxidized (OOA2) factors. OOA1 factor contained abundant
oxygen-containing fragments, accounting for more than 50% of the total mass spectrum, with an O/C of 0.42. In particular,
oxygenated fragment containing one oxygen accounted for 39%. The $C_2H_3O^+$ ion ($m/z$ 43) is an important component of less
oxidized SOA (Wang et al., 2020a), which was highly correlated with OOA1 (r = 0.94). OOA1 showed lower concentration
during daytime and higher concentration during nighttime (Fig. 2k). Such diurnal variation was similar to that of RH, but
opposite to that of solar radiation, which was similar to the previous report (Sun et al., 2014). OOA1 may thus be associated
with aqueous reactions when high RH. The co-occurrence of nighttime peak of OOA1 and COA suggested the primary
source of cooking emission may have considerably contributed to the production of OOA1.
The factor OOA2 had the highest O/C of 0.72 and contained 61% oxygen-containing fragments (Fig. 2d), which is very
similar to the spectra of OOA factor resolved in other cities (Aiken et al., 2009; Hayes et al., 2013). This factor was
correlated strongly with the fragment of $CO_2^+$ (r=0.79). Compared to OOA1, this factor showed obvious diurnal variations
with a major enhancement of around noontime (10:00-15:00), up to 2.9 µg m$^{-3}$, indicating photochemical production of SOA
during daytime. The variation of OOA2 also correlated with odd oxygen ($O_x=O_3+NO_2$). These features agree well with the
previous observation of characteristic of more oxidized OA (Hu et al., 2016).



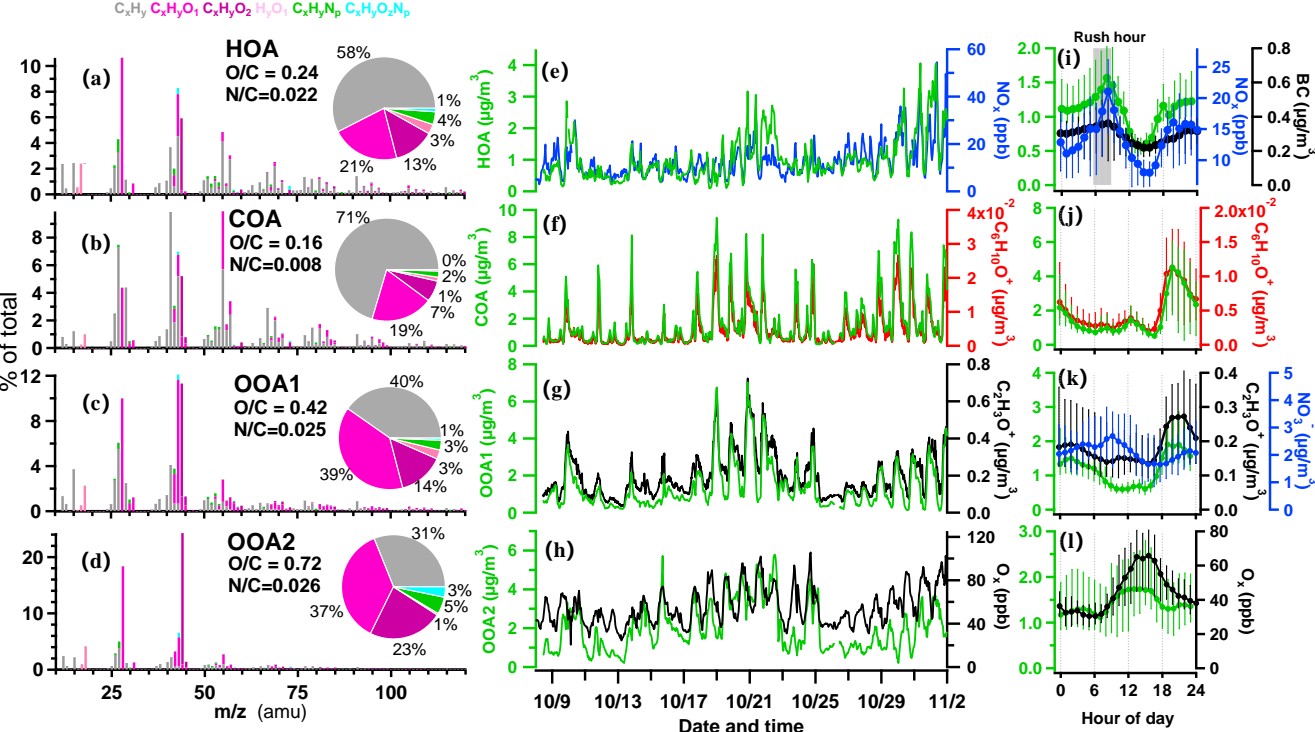

**Figure 2: Results of source-attributed organic aerosols by the PMF analysis on the OA mass spectra measured by the HR-TOF-AMS. Mass spectra of PMF factors for (a) hydrocarbon-like OA (HOA); (b) cooking OA (COA); (c) oxygenated OA1 (OOA1); (d) oxygenated OA2 (OOA2). The pies show the relative contributions of the six ion categories to each factor. (e-h) Temporal variations of four OA factors (HOA, COA, OOA1 and OOA2) with each correlated species as $NO_x$, $C_6H_{10}O^+$, $C_2H_3O^+$ and $O_x$. (i-l) Diurnal profiles of $NO_x$, BC, $C_6H_{10}O^+$, $C_2H_3O^+$, $NO_3^-$ and $O_x$. The lines and whiskers denote the median, the 25th and 75th percentiles at each hour, respectively.**

### 3.3 Source attributed VOCs

Fig. 3 and Fig. S5 summarize the source attributed of VOCs and key indicators. The first factor is dominated by aromatic compounds, such as $C_6H_6$ (*m/z* 79.054) and $C_7H_8$ (*m/z* 93.070) in Fig. 3a, which are well established markers for vehicle emissions (Gkatzelis et al., 2021), with good correlations of this factor (r = 0.97 and 0.63, respectively Fig. S6). This factor showed peaks in the morning and afternoon rush-hour (Fig. 3k) and was well correlated with HOA and $NO_x$ (r = 0.63 and 0.53 Table S1), with a major increase during the early morning, reaching a peak value of 12.1 ± 1.3 ppb at 08:30, further corroborating the traffic source of this factor. The concentration decreased around noontime until 15:00 because of the dilution by well-developed boundary layer and its consumption through photochemical reaction due to intense solar radiation. The diurnal pattern was also high at night, although the peak was lower than early morning, the average concentration was higher by a factor of 2 than daytime. This suggests that traffic VOCs prefer to participate in photochemical reaction, and other primary emissions may be precursors during nocturnal chemistry.



The second factor contained abundant aldehydes such as $C_3H_4O$ (acrolein), $C_3H_6O_2$ (hydroxyacetone), $C_6H_{12}O$
(hexanal), and $C_7H_{14}O$ (heptanal) at m/z 57.069, 75.044, 101.096, and 115.112, respectively, as well as $C_8H_{10}$ (C8-aromatics)
and $C_9H_{12}$ (C9-aromatics) at m/z 107.086 and 121.101 in VOC mass spectra (Fig. 3b), which are footprint VOCs identified
from primary cooking emission during the charbroiling and frying (Klein et al., 2016). This factor had similar time series
(Fig. 3g) with COA and had high correlation (r=0.67, Table S1). The concentration of this factor decreased during the
daytime, and yet surged after 18:00 with a peak value at 19:00 (17.2 ± 3.0 ppb). As shown in Fig. 3l, the diurnal pattern
decreased strongly after emission throughout the night, suggesting that cooking VOCs may be major precursors and were
consumed during night.
Besides the two primary VOCs, three secondary oxygenated VOCs (SecVOCs) factors are identified. These three
factors are not well correlated with any primary factors attributed by the OA, are thus considered to be mainly contributed by
secondary production. Among these three factors, SecVOC2 contained isoprene ($C_5H_8$, at *m/z* 69.070), and also some first
generation oxidation product, methyl vinyl ketone and methacrolein (MVK+MACR, m/z 71.049), which were produced by
enhanced biogenic emission of vegetations under solar radiation (Jordan et al., 2009; Cheng et al., 2018; Zhang et al., 2020a;
Gu et al., 2021). This factor was thus also contributed by this primary source. SecVOC2 contributed a large fraction of
$C_2H_4O_2$, $C_3H_6O_3$ and $C_4H_6O_3$ (Fig. 3c), which were gas-phase carboxylic acids, the oxidation products from photochemical
processes (Hartikainen et al., 2018; Li et al., 2021b). SecVOC2 had one clear peak at 13:00 (14.7 ± 2.8 ppb Fig. 3m). The
time series of SecVOC2 correlated well with OOA2 (r = 0.67), following the variation of solar radiation. It had a rapid
enhancement starting from 07:00 to 12:00 and declined continuously after 13:00. This indicated that many of these species
can be formed rapidly during daytime and may have a short lifetime owing to the partitioning to the condensed phase and
forming SOA.
SecVOC1 factor featured with some less-oxygenated VOCs e.g., $C_2H_2O_2$, $C_6H_{10}O_2$ and $C_{10}H_{14}O$ (Fig. 3d). This factor
had a peak at 19:00 which was consistent with the primary cooking VOCs factor, but also increased throughout the night,
peaking at midnight. This factor had a similar temporal trend with OOA1 (r = 0.76, Fig. 3i), which was less oxygenated than
photochemistry dominated OOA2. Combining the features above, SecVOC1 tended to be contributed by some immediately
reacted species from emissions in the late afternoon and early night. A particular factor (Fig. 3e) is significantly composed of
large molecular weight (large-MW) oxidized VOCs, i.e. the average on relative contribution of ionic compounds with
m/z>120 was above 50% in this factor (Fig. S5e), which was much higher than that in SecVOC2 (Fig. S5c). Fig. 3e shows
its signature compounds of $C_8H_{14}O$, $C_6H_{12}O_3$ and $C_8H_4O_3$, and some are nitrogen-containing VOCs, such as $C_6H_5NO_3$ and
$C_8H_9NO_3$. These VOCs with m/z>120 tend to be intermediate-volatility organic compounds (IVOCs) as the estimated vapor
saturation concentration is less than 6.5 μg m$^{-3}$ (Fig. S2). This factor is hereby termed as large-MW VOCs to indicate the
fraction of IVOCs, which only require few oxidation steps to become semi-volatile (Robinson et al., 2007). Fig. 3o shows an
increase of this factor at mid-night, later than the peak of SecVOC1, which may imply the ageing process in producing these
VOCs.

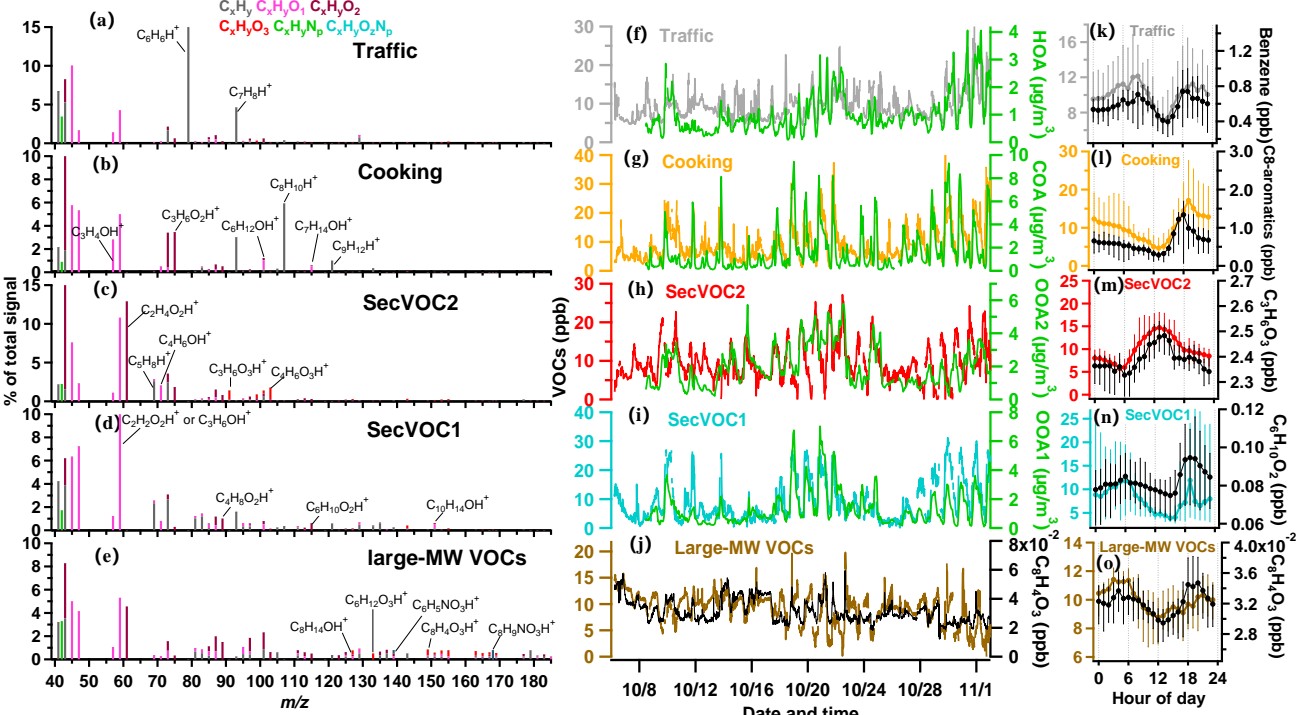

**Figure 3.** Source attributed VOCs by the PMF analysis measured by the PTR-TOF-MS. (a-e) Mass spectra of the five factors (traffic VOCs, cooking VOCs, secondary VOCs (SecVOC2, SecVOC1) and large molecular weight (MW) VOCs (large-MW VOCs), with major fingerprint peaks labelled in the mass spectra. (f-j) Temporal variations of the five VOCs factors and their respective correlated OA species. (k-o) Diurnal profiles of the VOC factors and their respective signature species. The lines and whiskers denote the median, the 25$^{th}$ and 75$^{th}$ percentiles at each hour, respectively.

## 3.4 Oxidation process of organics in the day and night

After source attribution of organics in aerosol and gas phase, we are able to identify the emission structure of primary sources and the consequent evolution. Given the identified primary traffic and cooking sources were emitted in the day and late afternoon respectively, this provides the potential opportunity to study the evolution of different primary sources in the potentially contrasting ageing mechanisms in the day and night.

Fig. 4a-d shows the temporal evolution of the key factors in a week to reflect the day and night ageing. The O/C ratio in the day is higher than that at night by 0.2 (Fig. 4a), corresponding with the increase of highly oxygenated fragments of AMS $f_{44}$ ($CO_2^+$) in the day (Fig. 4b), but increase of moderately oxygenated fragment $f_{43}$ ($C_2H_3O^+$) in the night. The diurnal variation of atomic ratio O/C of OA (Fig. 4e) showed a slight decrease at morning rush-hour and sharp drop at 18:00 (by 0.2) due to the significant contribution of the traffic and cooking sources respectively, corresponding with the two steps of increase at each time. This variation of O/C clearly showed the daytime and nighttime ageing processes of OA for different primary sources. Notably, the O/C showed peak value of 0.54 at 15:00, implying the importance of photooxidation in producing highly oxygenated OA.



Considering the diurnal pattern of anthropogenic activities, the traffic emission at rush-hour is deemed to be the major
source in contributing to the daytime production of OOA. The ratio OOA2/HOA is thus used to indicate the daytime
oxidation of OOA. As Fig. 4d and 4f shown, OOA2/HOA had a clear peak during daytime, and increased after 8:00 and
peaked at 14:00. After 15:00, the ratio gradually decreased to minimum at 20:00 and maintained to be low throughout the
night. This clearly demonstrated the photooxidation in producing OOA2 from oxidizing HOA. Fig. 4c and 4g give a few
examples of photooxidation in gas phase: the toluene showed a production of $C_3H_6O_3$ (hydroxypropionic acid) and $C_4H_6O_3$
(acetic anhydride) by a factor of 2 in 3 hours (Fig. S7c); the $C_3H_4O$ (acrolein) produced the oxidized product $C_2H_2O_2$
(glyoxal) by a factor of 1.3 (Fig. 4h). All of these reacted species are from traffic VOCs and the corresponding products are
from SecVOC2. The daytime biogenic emission, e.g. isoprene, may also contribute to the SecVOC2 formation by interacting
with OH in the presence of $NO_x$ (Lin et al., 2013), producing methacrylic acid epoxide (MAE) as the intermediate involved
in SOA formation, where a considerable nitrogen content was also found in OOA2 factor (N/C=0.026). Overall, the highly
oxidized OOA2 (O/C=0.72) is considered to be mainly contributed by traffic source, via oxidation of VOCs and partitioned
to condensed phase, direct oxidation on HOA through heterogenous oxidation (Guo et al., 2020), or VOCs evaporated from
HOA and further condensed after oxidation (Zhao et al., 2015). All factors may have contributed to the daytime production
of OOA2.
The nocturnal oxidation is mainly contributed by the sources emitting from late afternoon throughout the midnight.
Both traffic and cooking sources contributed to the emission since late afternoon, with cooking source as the predominant
contributor in both aerosol (Fig. 2j) and gas (Fig. 3l) phases. The ratio between the nighttime OOA1 and cooking aerosol
(OOA1/COA) is therefore used to indicate the nocturnal oxidation of SOA (Fig. 4d and j). There was a sudden increase of
OOA1/COA during the daytime because COA was consumed rapidly in the afternoon after a small amount of emission at
noon (Fig. 2j). The lowest OOA1/COA at 0.5 corresponded with the fresh cooking emission at 18:00, and kept increasing
until peaking in the early morning at 6:00 up to 3, which was an increase by a factor of 6 compared to the minima (Fig. 4j).
In addition to evidence for daytime reaction, Fig. 4h and Fig. S7d also gave evidence for certain reacted species during the
ageing at nighttime. The first-generation oxidation products from acrolein (ACR $C_3H_4OH^+$, m/z 57.033) are glyoxal
($C_2H_2O_2$, m/z 59.036) and formaldehyde ($HCHOH^+$, m/z 31.018) (according to the database in Master Chemical Mechanism
(MCM). This night oxidation is also evidenced by the formation of some nitrated organic compounds and ketone
compounds, such as $C_6H_5NO_3$ (nitrophenol, NP, Fig. 4l) and $C_6H_{10}O_2$ (hexanedione, Fig. 4k) produced from phenol ($C_6H_6O$)
and hexanal ($C_6H_{12}O$), respectively. Notably, an important fraction of large molecular weight-VOCs (which are mostly
intermediate VOCs, Fig. S2) peaked at 3:00 - 4:00 (Fig. 3o), consistent with the variation of OOA1/COA during night. Some
of these nitrated and oxygenated IVOCs may have been further oxidized and partitioned to aerosol phase, contributing to the
OOA1. Given the larger molecular usually has a lower O/C ratio (Hatch et al., 2017) (because of a higher content of carbon),
this may explain the lower O/C observed for nighttime formed OOA1 (O/C=0.42), than OOA2 produced by daytime



photooxidation (O/C=0.72). Notably, nighttime SOA had a high N/C (0.025), implying the $NO_3\cdot$-initiated from cooking
emissions oxidation, which was different from the organic nitrate formation mechanism during the daytime.

**Figure 4. Time series showing the aging of aerosols and gases, with grey vertical bars denoting nighttime (00:00-06:00), (a) the oxygen to carbon ratio (O/C); (b) the fragment fraction at *m/z* 44 ($f_{44}$), m/z 43 ($f_{43}$); (c) concentration ratio of $C_6H_{10}O_2$ to $C_6H_{12}O$ (hexanedione to hexanal), glyoxal to acrolein, and formaldehyde to acrolein; (d) concentration ratio of OOA1 to COA and OOA2 to HOA. Diurnal variations of key species showing the daytime photooxidation and nighttime oxidation, (e) ratio of O/C; (f) ratio of OOA2 to HOA; (g) concentration of $C_7H_8$ and ratio of $C_3H_6O_3$ to $C_7H_8$ (hydroxypropionic acid to toluene); (h) concentration of $C_3H_4O$ and ratio of $C_2H_2O_2$ to $C_3H_4O$ (glyoxal to acrolein); (i) particle diameter ($D_P$); (j) ratio of OOA1 to COA; (k) concentration of $C_6H_{12}O$ and ratio of $C_6H_{10}O_2$ to $C_6H_{12}O$ (hexanedione to hexanal); (l) concentration of $C_6H_6O$ and ratio of $C_6H_5NO_3$ to $C_6H_6O$ (nitrophenol to phenol).**





Fig. 5 summarizes the key indicators for the contrasting daytime and nighttime oxidation process. The species are
normalized by ΔCO,  where ΔCO is the total CO concentration subtracted by the background concentration (1[th] percentile of
the dataset), to indicate the variation of species regardless of the boundary layer evolution (Gouw and Jimenez, 2009). The
odd oxygen $O_x$ ($O_3$+$NO_2$) has been widely used to generally indicate the activity of daytime photochemistry (Hu et al.,
2017), and the enhanced moisture is main driving factor for nighttime chemistry. The $O_x$ concentration and RH are therefore
used as references with which the variations of species were correlated in the day and night, respectively. As Fig. 5a shown,
the traffic primary emissions in both gas (traffic VOCs) and aerosol phase (HOA) declined with increased $O_x$ by 60 % and
40% respectively, suggesting their roles as precursors in the daytime reaction. The produced species are oxygenated
SecVOC2 and OOA2, showing enhancement with $O_x$, peaking at midday-afternoon. This process was rapid as the SOA
production by a factor of 2.5 and the oxygenated VOCs production by a factor of 1.7 within 6 hours. The traffic VOCs are
widely observed to contribute to SOA production, with aromatic compounds serving as key precursors (Fang et al., 2021).
The semi-volatile nature of HOA means it could be evaporated to gas phase and further oxidized to recondense as SOA
(Robinson et al., 2007). The decrease rate of HOA with increased photochemical age was also found in urban environment
(Zhu et al., 2021), generally consistent with the reacted rate in this study. Here we linked the declining rate as a function of
photochemical activities for both reacted aerosol and gas phases for traffic sources. The gases evaporated from aerosol phase
(especially under higher temperature) and primary VOCs may be simultaneously involved in the photooxidation, further
contributing to the SOA formation.
For nocturnal oxidation shown in Fig. 5b, the reacted species are cooking VOCs and COA (decrease by 35 % and 77 %,
respectively), producing SecVOC1 and large-MW oxygenated VOCs, with an increase of night SOA formation by a factor
of 1.7. The nocturnal processes may have largely involved aqueous reactions, because the variations of reacted or produced
species were highly correlated with RH (Fig. 5b). The large-MW VOCs (mostly IVOCs) increased by 50% and reached
maxima when highest RH. This suggests the moisture may have been involved in converting some primary VOCs to IVOCs,
which further contribute to the SOA production during nighttime. Previous studies also found the oxidation of IVOCs  from
cooking sources can be an important source of SOA (Zhang et al., 2020b). The evidence is given here that organic aerosols
and gases from cooking emission had been reacted and contributed to SOA. The production rate of 0.2 µg m$^{-3}$ h$^{-1}$ was
generally consistent with previous laboratory work using gas precursors from cooking sources (0.07-0.5 µg m$^{-3}$ h$^{-1}$) (Liu et
al., 2017). Notably, daytime SOA had a higher oxidation state, implying the importance of photooxidation in producing
highly oxidized OA. This may be because of the high temperature at daytime and a species may require a lower volatility
(hereby more oxygenated) to be in condensed phase than at night.



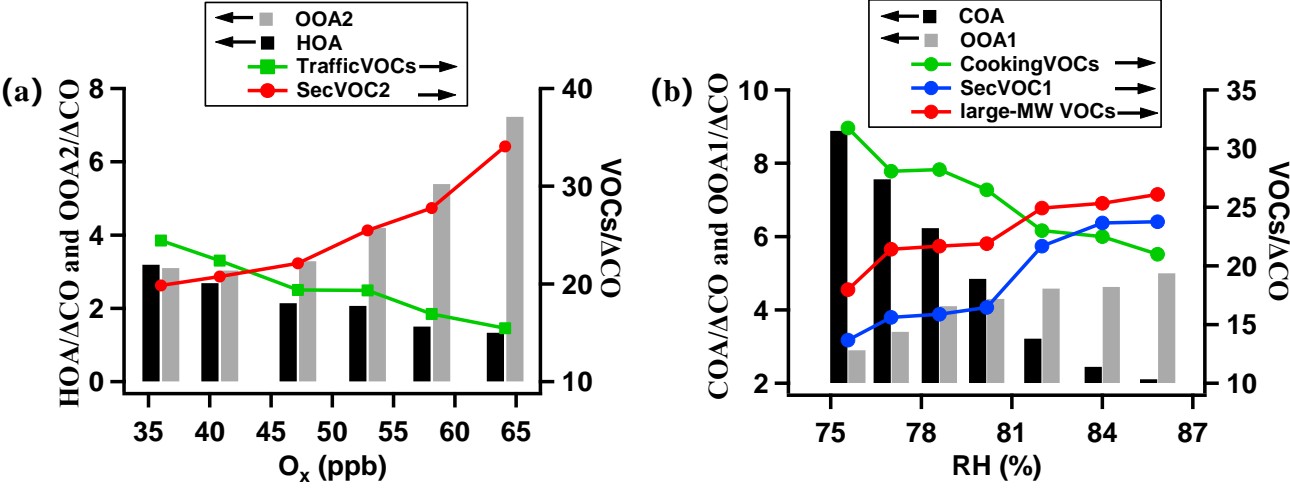

Figure 5. Daytime and nighttime evolution of key species against $O_x$ and RH respectively. (a) Daytime oxidation showing the primary emissions of traffic source in gas and aerosol phases including HOA, traffic VOCs; and secondary products including SecVOC2 and OOA2, with concentrations all normalized by $\Delta CO$. (b) Nighttime oxidation showing the primary emissions of cooking source in gas and aerosol phases including COA and cooking VOCs; and secondary products including SecVOC1, large-MW VOC and OOA1, with concentrations all normalized by $\Delta CO$.

## 4 Conclusion

In this study, organic gases and aerosols were concurrently characterized through online mass spectrometers at a megacity in central China. Through the factorization analysis on the organic mass spectra, two principal sources - the traffic and cooking sources were identified for both aerosol and gas phases, hereby the reacted and produced species between phases were interlinked. We observed clear evidence of daytime and nighttime oxidation of source-attributed OA and VOCs. Daytime photooxidation caused 60 % decrease of primary aerosol and 40 % of primary VOCs reduction for traffic sources, producing oxygenated SecVOCs by a factor of 1.7 and OOA by a factor of 2.5, in a 6 hours photochemical ageing. Nocturnal ageing caused a reduction of primary OA (by 77%) and primary VOCs (by 35%) from cooking sources, producing oxygenated VOCs and OOA by a factor of 1.4 and 1.7, respectively. In particular, larger molecular IVOCs produced (by a factor of 1.7) at night may importantly contribute to the OOA. This implies primary species in aerosol and gas phases both contribute to the production of OOA. A higher oxidation state of OOA from daytime photooxidation was found than nighttime, suggesting different compositions of produced OOA modulated by solar radiation and moisture, respectively. As vehicle and cooking emissions are the major contributors of organic aerosols in urban areas, especially in megacities. These results provide direct observations about the reaction rate for primary precursors and production rate for secondary aerosols, as influenced by primary sources and meteorological conditions. The environmental policy making should therefore consider the respective primary sources and ageing mechanisms for local and regional atmospheric environmental problems.

## Data availability

The data in this study are available from the corresponding author upon request.



**Author contribution**

DL, SK, and SL led and designed the study. SL, YW, HZ, YC, SZ and DH set up and conducted the experiment. SL, DL, YW, KH, XJ and SD contributed to the data analysis. QL, DZ, JS provided technical support and assistance. SL wrote the manuscript draft. DL and SK provided critical review and substantially revised the manuscript. All authors read and approved the final manuscript.

**Competing interests**

The authors declare no competing interests.

**Acknowledgments**

This work was supported by the National Key R&D programme of China (2019YFC0214703), National Natural Science Foundation of China (Grant Nos. 42175116, 41875167).

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
