# Peer review of "Evolution of source attributed organic aerosols and gases in a megacity of central China"

_Atmospheric Chemistry and Physics, 2022_

## Author Comment (AC1)

**Reviewer 1**

This study presented concurrent measurements of detailed mass spectra of aerosol particles and gases in an urban environment in Wuhan, a megacity in central China. The authors emphasized the two most intensive anthropogenic sources, the traffic source and cooking, and in details investigated the primary emission, secondary production/transformation of organic aerosols and gases, under the influence of two typical meteorological conditions. The results and analysis provide insights and improve the understanding on the organic aerosol-gas conversion and SOA formation, for the important sources on high population exposure in a typical anthropogenically polluted environment. Because the concurrent measurements in both phases are rare, in particular for the region in central China, it is also a very valuable dataset. I would recommend for its publication after addressing the following minor comments.

We thank reviewer for the insightful and positive comments. We have now addressed all comments and revised our previous manuscript accordingly. Reviewer's comments are in black italics. Our reply is in red, and the corresponding changes in the texts are highlighted in yellow.

*1) It will be useful to add some diagnosis for the PMF results from measured VOCs. More discussions on the choice of PMF factors should be given for VOCs. The conventional PMF analysis on VOCs only used a few species. Here a bunch of species are used. The difference regarding the species input for PMF should be given, and what is the advantages and disadvantages between both approaches.*

We thank the reviewer to point this out. The detailed diagnosis for the PMF factors from the PTR measurement is now added in Text S2 and Figure S4.

The discussions regarding the different method using PTR measurements compared to conventional GC-MS measurements on the VOCs are added.

"Here 109 species of VOCs from the PTR-TOF-MS were used, which are mostly oxygenated or contain cyclic functional groups, in contrast with the GC-MS measurement which contains many aliphatic hydrocarbon compounds and usually only decades of species were used for the PMF analysis (Zheng et al., 2021). These may lead to some unsolved primary VOC sources for the current analysis using the PTR-measured VOCs, if the source contained crucial markers of aliphatic species, but this method may have great advantage in comprehensively resolving factors of secondary VOCs. This may explain the less primary but more secondary VOC factors compared to conventional PMF-resolved VOC sources (Cai et al., 2010)."

Line 210-216.

*2) It is interesting to see a substantial fraction of secondary oxygenated VOCs and even for oVOCs, a few factors can be resolved with different levels of oxidation. This reflects the feature of PTR measurement using proton as ion source, which prefers to measure oVOCs. Some discussions will need to discuss the nature of oVOCs you observed rather than primary hydrocarbons, which may place some limitation for the conclusion here.*

The limitation using PTR-measured VOCs for the PMF analysis is now discussed as the answer to the above question.

*3) It would be useful to point out what the policy maker could benefit from current study, such as the potential benefits to regulate the nighttime and daytime emissions. The importance in*

*controlling the highly oxidized species in the daytime may be emphasized. This will help increase the impacts of this study.*

The related discussions are now added according to reviewer's suggestions.

"The particular regulation should be placed to avoid the formation of daytime highly-oxidized species when high solar radiation, which may contribute to the reactive oxygen species and exert adverse health impacts (Tao et al., 2003; Verma et al., 2009)."

Line 377-379.

Other technical correction:

*Line 26, needing to define O/C*

It has been revised.

"...in a higher oxidation state (oxygen-to-carbon, O/C = 0.72)…"

Line 26-27.

*Line 41, before? Needing changing a word.*

It has been revised.

"This raises challenges for source-oriented environmental policy making, until explicit understanding on the formation mechanism of SOA from different sources."

Line 41.

*Line 42, allow the online attribution*

It has been revised.

"The application of detailed mass spectra of organic aerosols allows the online source attribution of organic aerosol (OA)."

Line 42.

*Line 69, in a typical region of mixed sources of ...*

It has been revised.

*Line 74, initializing*

It has been revised.

*Line 75, The cluster analysis*

It has been revised.

*Line 102, VOC compound*

It has been revised.

*Line 103, at which temperature?*

We have now added temperature.

"The vapor saturation concentration (equilibrium vapor pressures) ($C_*$) of each VOC compound at 25 ºC is estimated using the parameterization based on elemental ratio and molecular weight."

Line 102-104.

*Line 121, the model fits the data well*
It has been revised.

*Line 149, high temperature*
It has been revised.

*Line 127, electrical mobility diameter*
It has been revised.

*Line 130, need a slope value for the PM closure*
We have added the slope value.
"…the SMPS agreed well with that from the sum of compositions by the HR-ToF-AMS and SP2 (r = 0.71, slope = 0.90)."
Line 131.

*Line 137, non-refractory*
It has been revised.

*Line 146, temporal evolution*
It has been revised.

*Line 150-154, what these different air mass histories can tell?*
Some discussions are added now.
"It therefore separates the periods with more local air mass or influences from regional transport."
Line 155-156.

*The quality of Fig. 1 needs to be improved.*
The figure has been improved.

*The y-axis are very confusing for Fig. 1d, is the green for OA?*
It has been revised.
"(d) mass concentrations of key aerosol species (the green left y-axis represents OA, the right y-axis represents other aerosol species);"
Line 159-160.

*Line 168, contained some fragment markers… may have been overwhelmed*
It has been revised.

*Line 173, maybe needing more decimal.*
It has been revised.
"This factor has a $C_3H_3O^+/C_3H_5O^+$ of 3.0 and $C_4H_7^+/C_4H_9^+$ of 2.0 (1.0 is usually for HOA),…"
Line 175-176.

*Line 174, marker fragment*
It has been revised.

*Line 174, a major peak*
It has been revised.

*Line 179, account for a mass fraction?*
It has been revised.
"Here the concentrations of COA in OA were in the range of 0.5-4.5 µg m$^{-3}$ and accounted for 23 % of OA mass fraction on average."
Line 181-182.

*Line 180, which was popular*
It has been revised.

*Line 181-182, needing to rewrite, as it already mentioned about the two factors, how could it be "further" separated.*
It has been rewritten.
"According to the oxidation state, OOA was separated into lower (OOA1) and more oxidized (OOA2) factors."
Line 184-185.

*Line 184, the oxygenated fragment*
It has been revised.

*Line 186, but higher concentration*
It has been revised.

*It needs some emphasis that OOA1 was highly associated with the RH.*
It has been revised.

*Line 191, similar to what factors in those studies? The discussions here need to be specific.*
We have added specific data in other cities.
"The factor OOA2 had the highest O/C of 0.72 and contained 61% oxygen-containing fragments (Fig. 2d), which is very similar to the spectra of OOA factor resolved in other cities (0.60 and 67% in Mexico City, 0.8 and 66% in Pasadena)..."
Line 193-194.

*Line 193, at round noontime of up to ..., indicating the photochemical production*
It has been revised.

*Line 195, the reference here needs to be specific, what kind of agreement.*
We have added specific data.

"The variation of OOA2 correlated with odd oxygen ($O_x = O_3 + NO_2$, r = 0.70), agreeing with previous observations that oxidized OA had strong correlation with $O_x$ (Wood et al., 2010; Hu et al., 2016)."
Line 197-199.

*Line 204-206, this sentence needs rewriting.*
It has been revised.
"The first factor was dominated by aromatic compounds, such as $C_6H_6$ (*m/z* 79.054) and $C_7H_8$ (*m/z* 93.070) in Fig. 3a. They were well established markers for vehicle emissions (Gkatzelis et al., 2021) and had a good correlation with this factor (r = 0.97 and 0.63, respectively Fig. S6)."
Line 217-219.

*Line 207-209, the sentence needs some breakups.*
It has been revised.
"This factor showed peaks in the morning and afternoon rush-hour (Fig. 3k) and was well correlated with HOA and $NO_x$ (r = 0.63 and 0.53 Table S1). The concentration of this factor had a major increase during the early morning, reaching a peak value of $12.1 \pm 1.3$ ppb at 08:30, further corroborating the traffic source of this factor."
Line 219-222.

*Line 210, in addition to its consumption*
It has been revised.

*Line 212, not preferred, is more likely*
It has been revised.

*Line 213, for nocturnal chemistry*
It has been revised.

*Line 217, cooking emissions*
It has been revised.

*Line 219, surged?*
It has been revised.
"The concentration of this factor decreased during the daytime and increased after 18:00 with a peak value at 19:00 ($17.2 \pm 3.0$ ppb)."
Line 231-232.

*Line 220, this sentence is hard to understand.*
It has been revised.
"As shown in Fig. 3l, the diurnal concentration decreased strongly after emission and continued to decline throughout the night, suggesting that cooking VOCs may be major precursors and were consumed during night."
Line 232-234.

*Line 239, the mean contribution*
It has been revised.

*Line 243, the vapor pressure needs a reference temperature, is it in logarithmic?*
It has been revised.
"These VOCs with m/z>120 tend to be intermediate-volatility organic compounds (IVOCs) as the estimated vapor saturation concentration (log$_{10}$$C$*, 298 K) is less than 6.5 μg m$^{-3}$."
Line 254-255.

*Line 259, was. You need to check through the tense.*
Thank you for pointing out this problem. We have revised the tense of the full manuscript.

*Line 260, but increase?*
It has been revised.
"During night, the O/C ratio increased with the increase moderately oxygenated fragment $f_{43}$ (C$_2$H$_3$O$^+$)."
Line 273-274.

*Line 268, showed*
It has been revised.

*Line 275, as intermediately involved*
It has been revised.

*Line 276, in the OOA2 factor*
It has been revised.

*Line 277, by the traffic source, via the oxidation, partitioning, further condensation*
It has been revised.
"...is considered to be mainly contributed by the traffic source, via the oxidation of VOCs and partitioning to condensed phase, direct oxidation on HOA through heterogenous oxidation, or VOCs evaporated from HOA and further condensation after oxidation"
Line 290-292.

*Line 291, remove or add a bracket*
It has been revised.

*Line 310, the concentrations are normalized*
It has been revised.

*Line 314, is the main driving factor*
It has been revised.

*Line 315, showed*

It has been revised.

*Line 323, reaction rate*
It has been revised.

*Line 356, the different*
It has been revised.

*Line 357, remove as*
It has been revised.

**References**

Cai, C., Geng, F., Tie, X., Yu, Q., and An, J.: Characteristics and source apportionment of VOCs measured in Shanghai, China, Atmos. Environ., 44, 5005-5014, https://doi.org/10.1016/j.atmosenv.2010.07.059, 2010.

Gkatzelis, G. I., Coggon, M. M., McDonald, B. C., Peischl, J., Gilman, J. B., Aikin, K. C., Robinson, M. A., Canonaco, F., Prevot, A. S. H., Trainer, M., and Warneke, C.: Observations confirm that volatile chemical products are a major source of petrochemical emissions in US cities, Environ. Sci. Technol., 55, 4332-4343, https://doi.org/10.1021/acs.est.0c05471, 2021.

Hu, W., Hu, M., Hu, W., Jimenez, J., Yuan, B., Chen, W., Wang, M., Wu, Y., Chen, C., Wang, Z., Peng, J., Zeng, L., and Shao, M.: Chemical composition, sources, and aging process of submicron aerosols in Beijing: Contrast between summer and winter, Journal of Geophysical Research: Atmospheres, 121, 1955-1977, https://doi.org/10.1002/2015JD024020, 2016.

Tao, F., Gonzalez-Flecha, B., and Kobzik, L.: Reactive oxygen species in pulmonary inflammation by ambient particulates, Free Radical Bio. Med., 35, 327-340, https://doi.org/10.1016/S0891-5849(03)00280-6, 2003.

Verma, V., Ning, Z., Cho, A. K., Schauer, J. J., Shafer, M. M., and Sioutas, C.: Redox activity of urban quasi-ultrafine particles from primary and secondary sources, Atmos. Environ., 43, 6360-6368, https://doi.org/10.1016/j.atmosenv.2009.09.019, 2009.

Wood, E. C., Canagaratna, M. R., Herndon, S. C., Onasch, T. B., Kolb, C. E., Worsnop, D. R., Kroll, J. H., Knighton, W. B., Seila, R., Zavala, M., Molina, L. T., DeCarlo, P. F., Jimenez, J. L., Weinheimer, A. J., Knapp, D. J., Jobson, B. T., Stutz, J., Kuster, W. C., and Williams, E. J.: Investigation of the correlation between odd oxygen and secondary organic aerosol in Mexico City and Houston, Atmos. Chem. Phys., 10, 8947-8968, https://doi.org/10.5194/acp-10-8947-2010, 2010.

Zheng, H., Kong, S., Chen, N., Niu, Z., Zhang, Y., Jiang, S., Yan, Y., and Qi, S.: Source apportionment of volatile organic compounds: Implications to reactivity, ozone formation, and secondary organic aerosol potential, Atmos. Res., 249, 105344, https://doi.org/10.1016/j.atmosres.2020.105344, 2021.

---

## Author Comment (AC2)

**Reviewer 2**

This manuscript for the first time studied the sources and evolution of aerosols and VOCs synchronously with corresponding state-of-the-art instruments in a megacity of Central China. Most uniquely, unlike other events in previous studies, as the emission control measures of 7th CISM Military World Games stressed mainly on the industrial emissions, the vehicle and cooking emission dominated the sources of organic aerosols and VOCs. Taking this opportunity, the study clearly separated the two sources and identified their emission evolution with different mechanisms in daytime and nighttime respectively under the real ambient air. It is quite valuable and of great significance. I think it is a well designed and prepared paper, and can be accepted after the following questions answered.

We thank reviewer for the comprehensive and overall positive comments on our study. We have now addressed all comments and revised our previous manuscript accordingly. Reviewer's comments are in black italics. Our reply is in red, and the corresponding changes in the texts are highlighted in yellow.

*Line 71, are only industrial sources controlled? More detailed emission control information should be given.*

More information is now given.

"Due to the preparation and hosting of the 7th CISM Military World Games during the experimental period, the government implemented strict emission reduction measures, particularly for the industrial sources and heavy vehicle emissions in the main roads. The more localized pollution sources, such as traffic emission from smaller sizes of vehicles and cooking sources dominated the pollution in this study,…"
Line 70-73.

*Line 149, it is also due to low relative humidity as Figure 1 shown.*
It has been revised.

*Line 164, the corrected name of N.L. et al. should be given.*
It has been revised.

*Line 187, to a previous report*
It has been revised.

*Line 192, the RH at noontime (10-15) is still high around 60%, higher than those of northern cities. Is it possible that SOA be formed through aqueous oxidation?*
The related discussions are added.
"Considering the high RH during the experiment (> 60 %, Fig. 1g), OOA2 factor may have also experienced aqueous chemistry and showed good correlation with sulfate (r = 0.82)."
Line 199-200.

*Line 247, there is evidence that isoprene in Beijing can be from vehicle emission, in this study, can it be attributed to vehicle emission? For example, Gu et al., AE, Investigation on the urban ambient isoprene and its oxidation processes; Cheng et al., JES, Atmospheric*

*isoprene and monoterpenes in a typical urban area of Beijing.... The authors can give related discussions.*

The related discussions are added.

"The daytime biogenic emission, e.g. isoprene, may also contribute to the SecVOC2 formation by interacting with OH in the presence of $NO_x$ (Lin et al., 2013), producing methacrylic acid epoxide (MAE) and methacrolein (MACR) as intermediately involved in SOA formation (Gu et al., 2021)."
Line 237-239.

*Line 299, is there evidence that $NO_3$· can be formed from cooking emission or is it an important formation pathway or source?*

We have not directly measured $NO_3$·. The only estimate is the nighttime $NO_x$ may be partially from cooking sources, but this cannot be discriminated from nighttime traffic sources.

*Line 312, the scaling of CO can be also influenced by the wind speed, temperature, etc, not only boundary layer. The sentence should be corrected.*

It has been revised.

"...to indicate the variation of species regardless of the boundary layer evolution, wind speed, wind direction and temperature."
Line 325-326.

*Line 325-326, I suggest a quantitative law or conclusion should be given. For example, can the ranges of higher temperature be given through this study?*

This is revised as:

"The gases evaporated from aerosol phase (especially under higher temperature when increased saturation pressure for semi-volatile or intermediate volatile species) and primary VOCs may be simultaneously involved in the photooxidation, further contributing to the SOA formation."
Line 338-340.

*Figure 5b, I am not sure why 3% scale of RH was adopted for the data classification. Also from Figure 5b, I cannot find the correction coefficients.*

The scale is now revised as 5%. R is now added in Fig. 5.

[Figure]

*Line 334, how the production rate of 0.2 µg m$^{-3}$ h$^{-1}$ be obtained. I can not find the calculation processes.*

This is now clarified:

"An approximate production rate of 0.2 µg m$^{-3}$ h$^{-1}$ of OOA from cooking source can be obtained by considering the ageing time of ~10 h (from COA peak 18:00 to OOA1 peak 4:00)."

Line 348-350.

**References**

Gu, C., Wang, S., Zhu, J., Wu, S., Duan, Y., Gao, S., and Zhou, B.: Investigation on the urban ambient isoprene and its oxidation processes, Atmos. Environ., 270, 118870, https://doi.org/10.1016/j.atmosenv.2021.118870, 2021.